# VCAM1/VLA4 interaction mediates Ly6C<sup>low</sup> monocyte recruitment to the brain in a TNFR signaling dependent manner during fungal infection

**Donglei Sun**[1����], **Mingshun Zhang**[1����¤a], **Peng Sun**[1], **Gongguan Liu**[1], **Ashley B. Strickland**[1], **Yanli Chen**[1], **Yong Fu**[1¤b], **Mohammed Yosri**[1,2], **Meiqing Shi**[1]*

**1** Division of Immunology, Virginia-Maryland College of Veterinary Medicine and Maryland Pathogen Research Institute, University of Maryland, College Park, Maryland, United States of America, **2** The Regional Center for Mycology and Biotechnology, Al-Azhar University, Cairo, Egypt

These authors contributed equally to this work.
¤a  Current address: Department of Immunology, The Key Laboratory of Antibody Technique of Ministry of Health, Nanjing Medical University, Nanjing, China
¤b  Current address: Academy of Animal Science and Veterinary Medicine, Qinghai University, Xining, China
* mshi@umd.edu

**Data Availability Statement:** All relevant data are within the manuscript and its Supporting Information files.

## Abstract

Monocytes exist in two major populations, termed Ly6C<sup>hi</sup> and Ly6C<sup>low</sup> monocytes. Compared to Ly6C<sup>hi</sup> monocytes, less is known about Ly6C<sup>low</sup> monocyte recruitment and mechanisms involved in the recruitment of this subset. Furthermore, the role of Ly6C<sup>low</sup> monocytes during infections is largely unknown. Here, using intravital microscopy, we demonstrate that Ly6C<sup>low</sup> monocytes are predominantly recruited to the brain vasculature following intravenous infection with *Cryptococcus neoformans*, a fungal pathogen causing meningoencephalitis. The recruitment depends primarily on the interaction of VCAM1 expressed on the brain endothelium with VLA4 expressed on Ly6C<sup>low</sup> monocytes. Furthermore, TNFR signaling is essential for the recruitment through enhancing VLA4 expression on Ly6C<sup>low</sup> monocytes. Interestingly, the recruited Ly6C<sup>low</sup> monocytes internalized *C. neoformans* and carried the organism while crawling on and adhering to the luminal wall of brain vasculature and migrating to the brain parenchyma. Our study reveals a substantial recruitment of Ly6C<sup>low</sup> monocytes to the brain and highlights important properties of this subset during infection.

## Author summary

Monocytes are white blood cells, circulating in the bloodstream and playing important roles during infections. There are two subsets of monocytes in mice: Ly6C<sup>hi</sup> and Ly6C<sup>low</sup> monocytes. In contrast to the recruitment of Ly6C<sup>hi</sup> monocytes shown in other infection models, we observed the predominant recruitment of Ly6C<sup>low</sup> monocytes to the brain post-capillary venules during intravenous infection with *C. neoformans*, a fungal pathogen

**Funding:** National Institutes of Health (NIH, https://www.nih.gov) provided funding to MS under grant number AI131219 and AI131905. The funders had no role in study design, data collection and analysis, decision to publish, or preparation of the manuscript.

causing brain infection. The recruitment is mainly mediated by the interaction of VCAM1 and VLA4, which are expressed on the brain endothelium and monocytes, respectively. We further demonstrate that TNFR signaling plays an essential role during Ly6C^{low} monocyte recruitment through enhancing VLA4 expression on monocytes. We also observed that Ly6C^{low} monocytes internalize *C. neoformans* and, together with the ingested organism, crawl along the luminal wall of brain vasculatures and migrate to the brain parenchyma. Thus, VCAM1/VLA4 interaction mediates Ly6C^{low} monocyte recruitment to the brain in a TNFR signaling dependent manner during fungal infection.

## Introduction

Derived from the bone marrow, monocytes are a heterogeneous population of leukocytes in the blood and play a central role during infection and inflammation [1, 2]. Two major subsets of monocytes have been defined based on the cell surface markers in mice: Ly6C^{hi} monocytes and Ly6C^{low} monocytes [2–4]. Ly6C^{hi} monocytes express high levels of Ly6C and CC chemokine receptor 2 (CCR2), but low level of CX3C chemokine receptor 1 (CX3CR1), while Ly6C^{low} monocytes express low levels of Ly6C and CCR2, but high level of CX3CR1 [2–4]. The corresponding populations in humans are CD14^{hi}CD16^- and CD14^{low}CD16^{hi} monocytes, respectively [5]. During infection and inflammation, Ly6C^{hi} monocytes are rapidly recruited to tissues in a CCR2-dependent manner [2, 6]. They secrete high levels of proinflammatory cytokines and differentiate into inflammatory dendritic cells and inflammatory macrophages, contributing to local and systemic inflammation [2]. In contrast, Ly6C^{low} monocytes have been shown to crawl along the endothelium of blood vessels in the dermis, mesentery, and kidney to scavenge microparticles from the luminal side in a steady state [7, 8], acting as luminal blood macrophages [1]. However, recent data suggest that Ly6C^{low} monocytes are also recruited to sites of inflammation, playing an anti-inflammatory role [9, 10] or proinflammatory role [11–13].

Monocyte recruitment is believed to be mediated by interactions of monocytes with the endothelium through adhesion moleculses [2]. Three types of adhesion molecules have been identified: the intergrins, the selectins, and the Ig superfamily members [14]. Adhesion molecules expressed by monocytes include L-selectin, PSGL1, CD11a, CD11b, and VLA4, while endothelial cells express P-selectin, E-selectin, ICAM1 and VCAM1 [2]. These molecules have been shown to mediate rolling, adhesion, or transmigration of monocytes in vitro, particularly using cultured monolayers of human umbilical vein endothelial cells [15–17]. However, relatively less is known about the adhesion molecules involved in monocyte recruitment in tissues, and the adhesion molecules for monocyte trafficking may differ in different types of tissues and different inflammation states [2]. Notably, the recruitment of Ly6C^{low} monocytes to the brain and the underlying mechanism during brain infections remains poorly understood. Understanding molecular mechanisms involved in monocyte recruitment would be of therapeutic interest for selective blocking of the cell trafficking.

*Cryptococcus neoformans* is an encapsulated pathogenic fungus, accounting for 180,000 deaths worldwide annually [18]. The infection initiates in the lung and the yeast cells can disseminate to the bloodstream, particularly in HIV patients due to impaired cellular immunity [19, 20]. Once the organisms enter the blood, they migrate to the brain across the blood-brain barrier, resulting in fatal brain infections [19, 20]. Thus, brain migration of *C. neoformans* is one of the critical steps for the disease progression. Although *C. neoformans* can enter the brain through direct invasion of the brain endothelial cells [21–25], evidence has been

provided that mononuclear phagocytes promote brain invasion of the organism [21, 26, 27]. *C. neoformans* can survive in monocytes [28, 29], and monocytes containing *C. neoformans* have been found present in the perivascular space of the brain [23, 27]. Intravenous administration of *C. neoformans*-infected macrophages can enhance the brain fungal burden [26], while depletion of monocytes can reduce the brain fungal burden [26, 27]. Notably, monocytes harboring *C. neoformans* have been directly seen to cross a monolayer of brain endothelial cells cultured in vitro [30, 31]. However, the dynamic interactions of monocytes with *C. neoformans* in the brain vasculature in vivo have not been elucidated, mainly because of the technical challenges of the in vivo imaging.

In this study, with the use of intravital microscopy (IVM) we demonstrated that Ly6C$^{low}$ monocytes were predominantly recruited to the brain vasculature using a model of acutely disseminated *C. neoformans* infection. The influx of Ly6C$^{low}$ monocytes was mainly mediated by the interaction of VCAM1 and VLA4. In addition, TNFR signaling was crucial for Ly6C$^{low}$ monocyte recruitment by enhancing its VLA4 expression. Finally, we showed that Ly6C$^{low}$ monocytes engulfed *C. neoformans* and carried the organism while crawling along the luminal wall of brain vessels and migrating to the brain parenchyma.

## Results

### Substantial recruitment of monocytes to the brain after infection with *C. neoformans*

Under resting conditions, the patrolling behavior of CX3CR1$^+$ monocytes has been visualized in the skin vasculature and mesentery vessels by IVM [7]. Similarly, we observed the crawling behavior of GFP$^+$ cells in the brain vasculature of naïve CX3CR1$^{gfp/+}$ mice through the craniotomy window using IVM (Fig 1A, S1 Video). The cells could crawl against the blood flow and exhibit a leading edge (front of the cells) (S1 Fig, S2 Video). Interestingly, we observed an impressive increase in GFP$^+$ cell recruitment to the brain vasculature after intravenous infection with *C. neoformans* (Fig 1B, S3 Video). The recruitment of GFP$^+$ cells was dramatically elevated 12 h post infection and maintained thereafter (Fig 1C). The increase of GFP$^+$ cells in the brain was confirmed by flow cytometry (Fig 1D). We further noticed that GFP$^+$ cells were recruited mainly in postcapillary venules but not precapillary arterioles (Fig 1E). Statistical analysis showed that the number of GFP$^+$ cells in postcapillary venules was significantly higher than the number of the cells in precapillary arterioles (Fig 1E). The recruitment of GFP$^+$ cells was almost completely abolished by depletion of monocytes using clodronate liposomes (S2 Fig), suggesting that the recruited GFP$^+$ cells were monocytes. Flow cytometry showed that > 95% of recruited GFP$^+$ cells expressed CD11b with a small number of cells expressing NK1.1, confirming that the vast majority of recruited GFP$^+$ cells were monocytes (Fig 1F & 1G). Thus, *C. neoformans* infection in the brain induced substantial recruitment of monocytes to the brain.

### Ly6C$^{low}$ monocytes are predominantly recruited to the brain

There are two subsets of monocytes in mice [2–4]. To characterize the phenotypes of recruited monocytes in the brain, we generated CX3CR1$^{gfp/+}$CCR2$^{rfp/+}$ mice and performed IVM on the brain of these mice 24 h following *C. neoformans* infection. The result indicated that the recruited monocytes expressed various levels of both CX3CR1 and CCR2 (Fig 2A). As it is difficult to clearly differentiate the two subsets based on levels of CX3CR1 and CCR2, we stained monocytes with Alexa Fluor 647 conjugated anti-Ly6C mAb and found that most monocytes expressed relatively low or even negative levels of Ly6C (Fig 2B). The result is consistent with

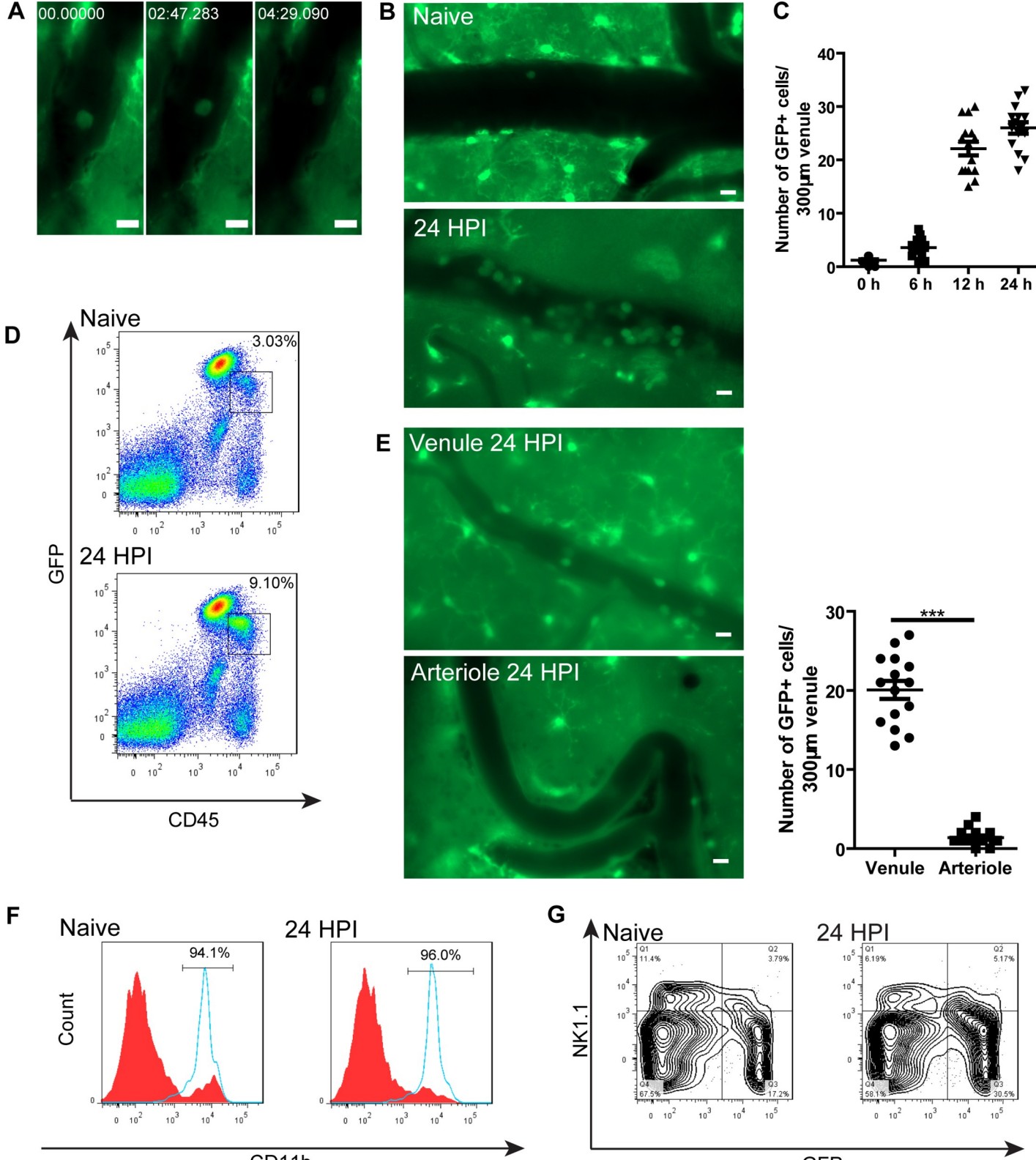

**Fig 1. Monocytes patrol the brain postcapillary venules under resting conditions and are substantially recruited after *C. neoformans* infection. (A)** A series of IVM images showing a GFP⁺ cell crawling inside a brain postcapillary venule of a naïve CX3CR1[gfp/+] mouse. See also S1 Video. **(B)** Representative IVM images showing a substantial recruitment of GFP⁺ cells to brain postcapillary venules of CX3CR1[gfp/+] mice 24 h after i.v. infection with 20x10⁶ *C. neoformans* as compared to naïve mice.

See also S3 Video. **(C)** The number of GFP$^+$ cells recruited to brain postcapillary venules of CX3CR1$^{gfp/+}$ mice (n = 5 per time point) at various time points after i.v. infection with 20x10$^6$ *C. neoformans*. The number of GFP$^+$ cells in each postcapillary venule of 300 μm in length were enumerated. **(D)** Representative flow cytometry plots of GFP$^+$ cells in the brain of naïve and infected CX3CR1$^{gfp/+}$ mice 24 h post infection with 20x10$^6$ *C. neoformans*. **(E)** The recruitment of GFP$^+$ cells in postcapillary venules comparing to precapillary arterioles in the brain of CX3CR1$^{gfp/+}$ mice (n = 5) 24 h post infection with *C. neoformans*. Left panel: representative IVM images; right panel: quantification. **(F)** Representative flow cytometry histograms showing CD11b expression on CD45$^+$GFP$^+$ cells (blue solid line) and CD45$^+$GFP$^-$ cells (red filled) isolated from the brain of naïve and infected CX3CR1$^{gfp/+}$ mice 24 h post infection with 20x10$^6$ *C. neoformans*. The percentages of CD11b$^+$ cells out of CD45$^+$GFP$^+$ cells were shown in the figure. **(G)** Representative flow cytometry plots showing NK1.1 expression on a small portion of GFP$^+$ cells in naïve and infected mice 24 h after infection. Initially, total CD45$^+$ brain leukocytes were gated. Scale bars: 10 μm. Data are expressed as mean ± SEM and representative of 2 independent experiments. $^{***}$, p<0.001.

flow cytometry data which showed that the Ly6C$^{low}$ subset comprised the majority of the CX3CR1$^+$ monocyte population recruited to the brain (Fig 2C & 2D). An increase in the Ly6C$^{hi}$ subset was also noticed but less pronounced compared to the Ly6C$^{low}$ population (Fig 2C & 2D). To further confirm the finding that the majority of recruited monocytes were Ly6C$^{low}$ monocytes, CCR2$^{rfp/rfp}$ mice (CCR2 was disrupted) were infected with *C. neoformans* to visualize monocyte recruitment, as CCR2 signaling is required for the Ly6C$^{hi}$, but not Ly6C$^{low}$, population to emigrate from the bone marrow [2, 6]. As expected, deficiency of CCR2 did not remarkably affect the influx of monocytes to the brain vasculature following *C. neoformans* infection as visualized by IVM (Fig 2E, S4 Video). In contrast to Ly6C$^{hi}$ monocytes whose recruitment was dramatically reduced in the absence of CCR2 signaling, Ly6C$^{low}$ monocyte recruitment was not affected 24 h post infection and only slightly affected 48 h post infection (Fig 2F). We speculate that this small difference noted at 48 h post infection may be attributed to the conversion of some Ly6C$^{hi}$ monocytes to Ly6C$^{low}$ monocytes as reported previously [32, 33]. In addition to monocyte recruitment, we detected an increase in NK cells (Fig 2D); however, most of these recruited NK cells did not express CX3CR1 in our experimental setting (S3B Fig). Interestingly, neutrophil recruitment was limited at 24 h post infection (Fig 2D) which was confirmed by IVM (S4 Fig); this is in sharp contrast to previous studies using LPS treatment [34]. Moreover, we found that the recruitment of Ly6C$^{low}$ monocytes is largely brain specific, as recruitments to other organs (including liver and kidney) are comparable to the systematic increase in the blood (S5 Fig). Finally, we confirmed that some of the recruited Ly6C$^{low}$ monocytes transmigrated to the brain parenchyma (S6 Fig). Taken together, these results demonstrate that Ly6C$^{low}$ monocytes are predominantly recruited to the brain vasculature during *C. neoformans* infection.

## Monocyte recruitment to the brain vasculature is mainly mediated by the VCAM1/VLA4 interaction

To study the molecular mechanisms involved in the recruitment of Ly6C$^{low}$ monocytes to the brain vasculature, we performed IVM on the brain of infected wild type mice and labeled monocytes using Alexa Fluor-647 conjugated anti-CX3CR1 mAb. CX3CR1$^+$ cells in the brain vasculature were stably and brightly labeled by the mAb (Fig 3A, S5 Video). Among various types of adhesion molecules, we first examined whether or not selectins are involved in the recruitment of monocytes during brain infection with *C. neoformans*. The result showed that blocking E-, P- and L- selectins had a negligible effect on CX3CR1$^+$ monocyte recruitment, suggesting that other molecules mediated the recruitment (Fig 3B). VCAM1 is another molecule capable of mediating rolling and adhesion [35, 36]; an increase in VCAM1 expression was detected in the brain after *C. neoformans* infection (Fig 3C). Importantly, treatment of infected mice with VCAM1 blocking mAb almost abolished monocyte recruitment to the brain vasculature (Fig 3D). Similarly, a blockade of VLA4 (VCAM1 ligand) also dramatically reduced monocyte recruitment to the brain vasculature (Fig 3D). S6 Video showed that VLA4 blocking

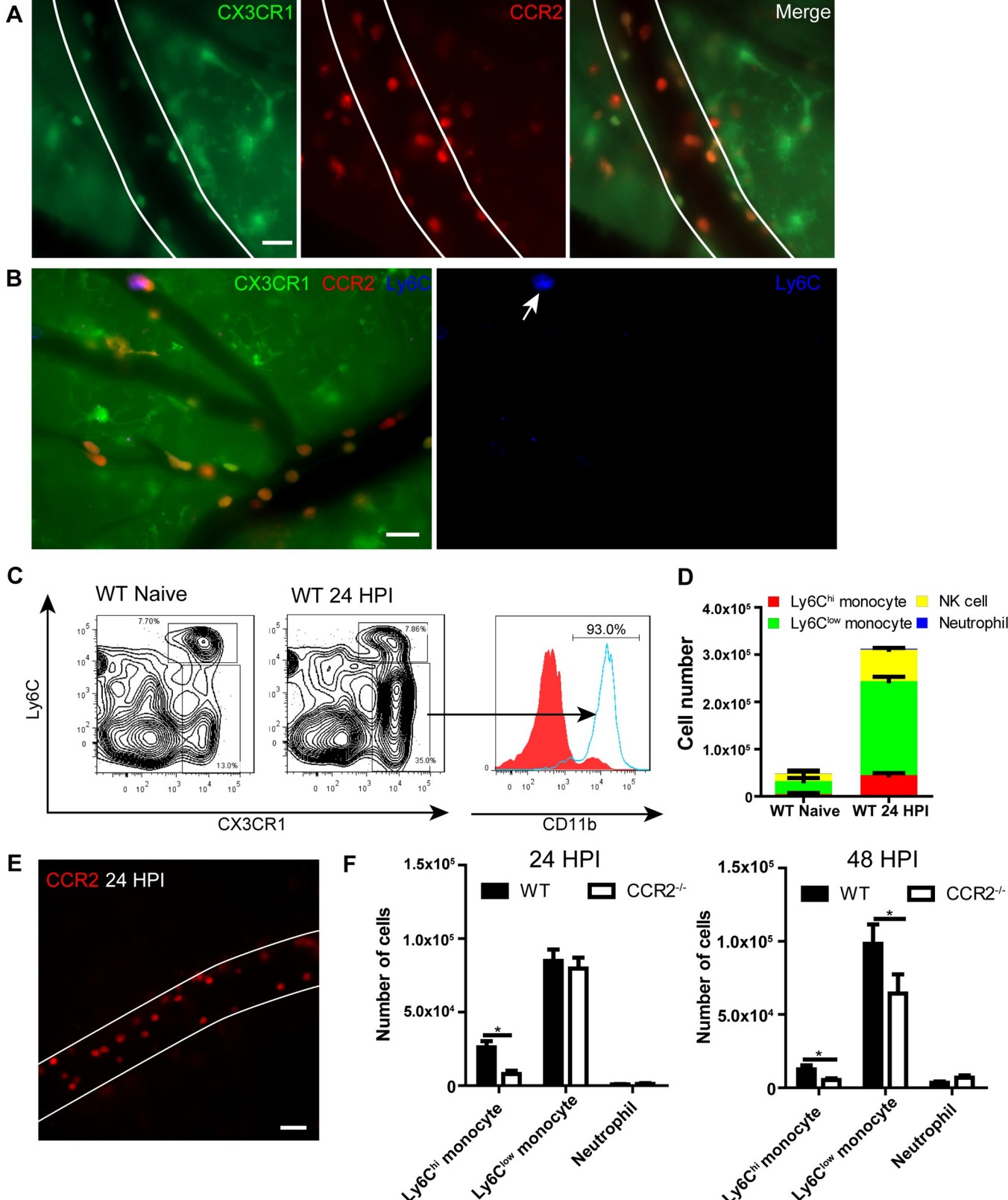

**Fig 2. Recruited monocytes are mainly Ly6C<sup>low</sup> populations.** (**A**) IVM images showing the expression of CX3CR1 and CCR2 on recruited monocytes in brain postcapillary venules of CX3CR1<sup>gfp/+</sup>CCR2<sup>rfp/+</sup> mice 24 h after infection with 20x10<sup>6</sup> *C. neoformans* (Green: CX3CR1, Red: CCR2). (**B**) Expression of

Ly6C on recruited monocytes (arrow) in brain postcapillary venules of CX3CR1$^{gfp/+}$CCR2$^{rfp/+}$ mice 24 h after infection with 20x10$^6$ *C. neoformans*. For labeling of Ly6C, mice were i.v. injected with 2 μg AF647-anti-Ly6C mAb through the tail vein 5 min before imaging. **(C)** Representative flow cytometry plots of CX3CR1$^+$Ly6C$^{hi}$ and CX3CR1$^+$Ly6C$^{low}$ monocyte subsets in the brain of naïve and infected C57BL/6 mice. Gated on CD45$^+$ cells. The expression of CD11b on CX3CR1$^-$ (filled) and CX3CR1$^+$ (solid) populations was shown on the right. C57BL/6 mice were i.v. infected with 20x10$^6$ *C. neoformans* and 24 h later brain leukocytes were isolated and analyzed by flow cytometry. **(D)** Flow cytometry analysis of the numbers of different subsets of cells recruited to the brain of C57BL/6 mice (n = 4 per group) 24 h after infection. Ly6C$^{hi}$ monocytes were defined as CD45$^+$Ly6G$^-$NK1.1$^-$CD11b$^+$CX3CR1$^+$Ly6C$^{hi}$, Ly6C$^{low}$ monocytes as CD45$^+$Ly6G$^-$NK1.1$^-$CD11b$^+$CX3CR1$^+$Ly6C$^{low}$, NK cells as CD45$^+$NK1.1$^+$, neutrophils as CD45$^+$CD11b$^+$Ly6G$^+$. **(E)** A representative IVM image showing the recruitment of RFP$^+$ monocytes in brain postcapillary venules of CCR2$^{rfp/rfp}$ mice (CCR2 deficient) 24 h after i.v. infection with 20x10$^6$ *C. neoformans*. See also S4 Video. **(F)** Flow cytometry analysis of the numbers of Ly6C$^{hi}$ and Ly6C$^{low}$ monocytes and neutrophils recruited to the brain of WT and CCR2$^{-/-}$ mice (n = 4 per group) 24 and 48 h post i.v. infection with 20x10$^6$ *C. neoformans*. Scale bars: 20 μm. Data are expressed as mean ± SEM and representative of 3 independent experiments. $^*$ $p<0.05$.

antibody efficiently wiped off the attached monocytes from the blood vessel and inhibited their rolling. A blockade of VCAM1 or VLA4 affected not only rolling but also adhesion of monocytes (Fig 3E). Flow cytometry confirmed that VCAM1 and VLA4 blocking was effective for both Ly6C$^{hi}$ and Ly6C$^{low}$ populations (Fig 3F). Collectively, these data demonstrate that the interaction of VCAM1 with VLA4 mediates recruitment of Ly6C$^{low}$ and Ly6C$^{hi}$ monocytes to the brain vasculature during *C. neoformans* infection.

## ICAM1 and CD11a but not CD11b are partially involved in monocyte recruitment to the brain vasculature

To study the involvement of β2 integrins in monocyte recruitment, we performed IVM on the brain of *C. neoformans*-infected wild-type mice, as well as ICAM-1$^{-/-}$, CD11a$^{-/-}$, and CD11b$^{-/-}$ mice. A great number of monocytes were observed in the postcapillary venules of infected wild-type and all three types of knockout mice (Fig 4A). A tendency of reduction and a slight but significant drop in the number of recruited monocytes were detected in infected ICAM1$^{-/-}$ and CD11a$^{-/-}$ mice respectively; however, monocyte recruitment in infected CD11b$^{-/-}$ mice was comparable to infected wild-type mice (Fig 4B). Of note, a significant reduction was detected in the percentage of adherent monocytes in the brain of infected ICAM1$^{-/-}$ and CD11a$^{-/-}$ mice (Fig 4C). S7 Video demonstrated the behavior of monocytes in CD11a$^{-/-}$ mice; most monocytes underwent fast rolling with very few cells adhering to the vessel, suggesting a role for CD11a in monocyte adhesion. Flow cytometry confirmed that there was significantly fewer Ly6C$^{low}$ monocytes recruited to the brain in ICAM1$^{-/-}$ and CD11a$^{-/-}$ mice 48 h post infection compared to wild-type mice (Fig 4D).

## TNFR signaling is required for monocyte recruitment to the brain vasculature via enhancing VLA4 expression on monocytes

TNFR signaling plays an important role in the recruitment of leukocytes [34, 37, 38]. Increase in TNF-α mRNA expression was detected in the brain of *C. neoformans*-infected mice (Fig 5A). Infected TNFR$^{-/-}$ mice exhibited dramatically reduced monocyte recruitment to the brain vasculature (Fig 5B, S8 Video). Flow cytometry confirmed that deficiency of TNFR led to a significant drop in the numbers of Ly6C$^{hi}$ and Ly6C$^{low}$ monocytes in the brain of infected mice (Fig 5C). Thus, TNFR signaling was critically involved in monocyte recruitment to the brain vasculature. Next, we examined how TNFR signaling affects monocyte recruitment during brain infection with *C. neoformans*. The results indicated that infection with *C. neoformans* led to an increase of VCAM1 expression on endothelial cells (Fig 5D) in both wild-type and TNFR$^{-/-}$ mice (Fig 5E). In contrast, *C. neoformans* infection induced a dramatic elevation of VLA4 expression on monocytes in wild-type mice but only a slight increase in TNFR$^{-/-}$ mice, resulting in a significantly higher expression of VLA4 on monocytes of infected wild-type mice compared to TNFR$^{-/-}$ mice (Fig 5F). This result suggested that TNFR signaling on

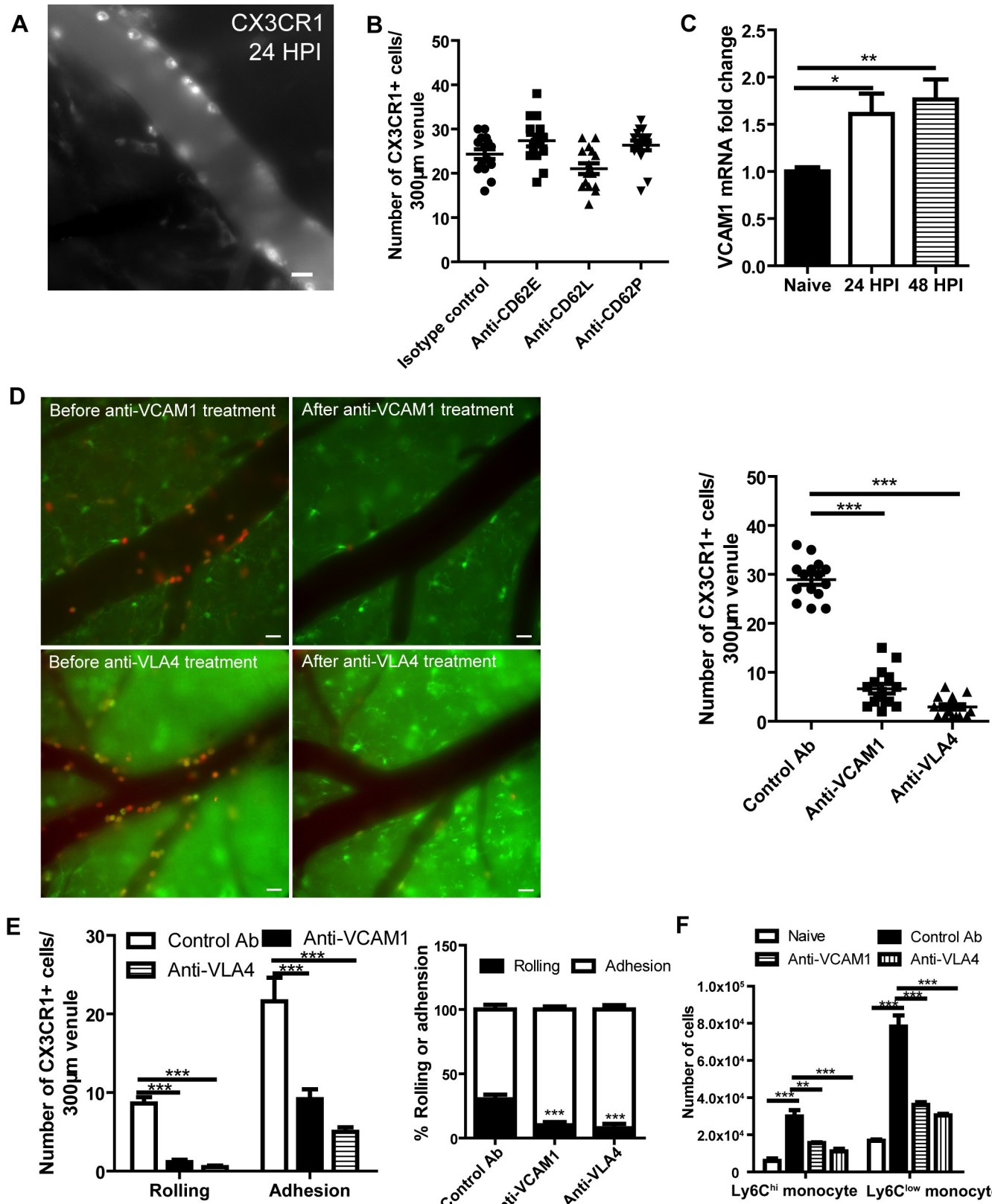

**Fig 3. VCAM1 and VLA4 interaction mediates the recruitment of monocytes to the brain vasculature. (A)** A representative IVM image showing the labeling of monocytes by anti-CX3CR1 mAb in brain postcapillary venules of C57BL/6 mice 24 h after i.v. infection with 20x10⁶ *C. neoformans*. Mice were i.v. injected with 2 µg AF647-anti-CX3CR1 mAb to label monocytes 5 min before imaging. **(B)** IVM analysis of the effect of blocking selectins on CX3CR1⁺ monocyte recruitment to brain postcapillary venules of C57BL/6 mice (n = 5 per group) 24 h after infection with 20x10⁶ *C. neoformans*. Mice were i.v. injected with 100 µg selectin-blocking mAbs or control Ab and 2 µg AF647 conjugated anti-CX3CR1 mAb (to label monocytes) 20 min and 5 min, respectively, before imaging. **(C)** The quantification of VCAM1 mRNA expression in the brain of mice (n = 5 per time point) before and after infection with *C. neoformans* using quantitative PCR. **(D)** Evaluation of the effect of anti-VCAM1 and anti-VLA4 mAbs on the number of total recruited CX3CR1⁺ monocytes to brain postcapillary venules of mice 24 h after i.v. infection with 20x10⁶ *C. neoformans*. Left panel: representative images of infected CX3CR1^gfp/+CCR2^rfp/+ mice showing monocyte recruitment before and after i.v. injection with 100 µg anti-VCAM1 or anti-VLA4 mAbs. Right panel: infected WT mice (n = 5 per group) were i.v. injected with 100 µg anti-VCAM1 mAb, anti-VLA4 mAb or control Ab 20 min before imaging. **(E)** IVM determination of the number and percentage of CX3CR1⁺ monocytes rolling on and adhering to brain postcapillary venules of C57BL/6 mice (n = 5 per group) 24 h after infection. Infected mice were i.v. injected with 100 µg anti-VCAM1 blocking mAb, anti-VLA4 blocking mAb or control Ab 20 min before imaging. **(F)** Flow cytometry determination of the numbers of Ly6C^hi and Ly6C^low monocytes in the brain of naïve and infected C57BL/6 mice (n = 4 per group). The infected mice were euthanized 24 h post infection. Infected mice were i.v. injected with 100 µg anti-VCAM1 blocking mAb, anti-VLA4 blocking mAb, or control Ab 20 min before euthanasia. Scale bars: 10 µm. Data are expressed as mean ± SEM and representative of 2 independent experiments. *, p<0.05, **, p<0.01, ***, p<0.001.

monocytes regulated the recruitment of the cells. To confirm this result, we purified bone marrow monocytes from wild-type and TNFR⁻/⁻ mice (both CD45.2 background) using negative selection method and labeled the cells with lipophilic dye CellVue and PKH26 respectively. The cells were mixed at 1:1 ratio and adoptively transferred into congenic CD45.1 recipient mice. The recipient mice were then i.v. infected with 20x10⁶ *C. neoformans* and euthanized 24 later to analyze brain monocyte recruitment by flow cytometry. The result showed that TNFR⁻/⁻ monocytes were recruited less compared to wild-type monocytes when equal numbers of monocytes were transferred during *C. neoformans* infection (Fig 5G). Collectively, TNFR signaling is critically involved in monocyte recruitment to the brain vasculature through enhancing VLA4 expression on monocytes during *C. neoformans* infection.

## Ly6C^low monocytes engulf *C. neoformans* and carry the organism while crawling on and adhering to the vessel wall and migrating to the brain parenchyma

Having addressed the mechanism underlying monocyte recruitment to the brain vasculature, we next directly visualized the dynamic interactions of recruited monocytes with *C. neoformans* and the vessel wall in the brain vasculature of CX3CR1^gfp/+ mice in real-time using IVM. The results showed that *C. neoformans* was engulfed by CX3CR1⁺ monocytes and monocytes carrying *C. neoformans* were frequently seen to crawl on postcapillary venules (Fig 6A, S7A Fig, S9 Video). In addition, CX3CR1⁺ monocytes carrying *C. neoformans* were observed to firmly adhere to the luminal side of postcapillary venules (Fig 6B, S10 Video). We also detected CX3CR1⁺ monocytes carrying *C. neoformans* which were in the process of crossing the vessel wall or were located in the brain parenchyma close to postcapillary venules (Fig 6C), supporting a role of monocytes in the transport of *C. neoformans* from the circulation to the brain parenchyma. Next, we examined the interactions of monocytes with *C. neoformans* in brain capillaries. In contrast to postcapillary venules, we found fewer CX3CR1⁺ monocytes in capillaries; the monocytes crawled slowly in capillaries (S11 Video) and adopted a rod-shaped morphology when passing through (Fig 6D, S11 Video). As shown in Fig 6D, monocytes in close proximity to, attaching to and engulfing *C. neoformans* were seen in capillaries. In addition to a single organism inside monocytes (Fig 6D), we also observed multiple yeasts within one monocyte (S7B Fig). As shown in S7B Fig, one of the three yeasts within the monocyte appeared to spread from the monocyte to an endothelial cell, a phenomenon of cryptococcal cell-to-cell spread described in vitro [28, 29]. Finally, we examined the phenotypes of monocytes carrying *C. neoformans*. Interestingly, only Ly6C^low monocytes were detected to carry *C.*

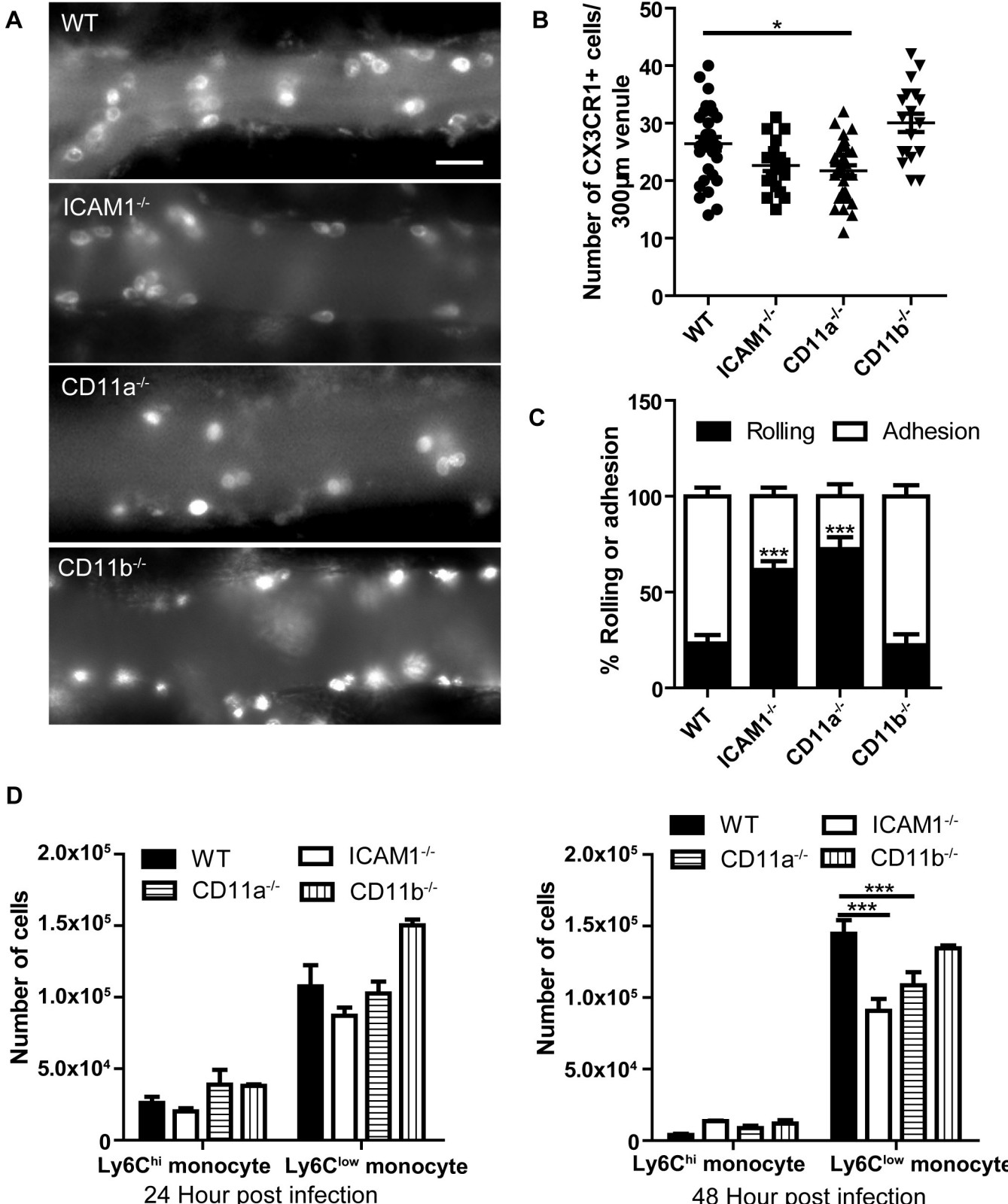

**Fig 4. ICAM1 and CD11a but not CD11b are partially involved in monocyte recruitment to the brain vasculature. (A)** Representative IVM images showing the recruitment of CX3CR1<sup>+</sup> monocytes in brain postcapillary venules of WT, ICAM1<sup>-/-</sup>, CD11a<sup>-/-</sup>, and CD11b<sup>-/-</sup> mice 24 h after i.v. infection with

20x10$^6$ *C. neoformans*. The infected mice were i.v. injected with 2 μg AF647-anti-CX3CR1 mAb to label monocytes 5 min before imaging. **(B)** IVM quantification of CX3CR1$^+$ monocytes recruited to brain postcapillary venules of WT, ICAM1$^{-/-}$, CD11a$^{-/-}$, and CD11b$^{-/-}$ mice (n = 5 per group) 24 h after infection. **(C)** IVM determination of the percentage of CX3CR1$^+$ monocytes rolling on and adhering to brain postcapillary venules of WT, ICAM1$^{-/-}$, CD11a$^{-/-}$, and CD11b$^{-/-}$ mice (n = 5 per group) 24 h after infection. **(D)** Flow cytometry determination of the numbers of Ly6C$^{hi}$ and Ly6C$^{low}$ monocytes recruited to the brain of WT, ICAM1$^{-/-}$, CD11a$^{-/-}$, and CD11b$^{-/-}$ mice (n = 4 per group) 24 h and 48 h after infection. Scale bar: 10 μm. Data are expressed as mean ± SEM and representative of 2 independent experiments. $^*$, $p<0.05$, $^{***}$, $p<0.001$.

*neoformans*, although Ly6C$^{hi}$ monocytes were also present in the postcapillary venules (Fig 6E). Flow cytometry confirmed that monocytes carrying *C. neoformans* were almost exclusively Ly6C$^{low}$ subset in the brain of infected CX3CR1$^{gfp/+}$ mice (Fig 6F, S7C Fig). It is important to point out that that the majority of yeast cells in the brain were outside phagocytes (S7C Fig). Taken together, these results demonstrate that Ly6C$^{low}$ monocytes internalize *C. neoformans* and bring the organism to the brain parenchyma.

## Discussion

Monocytes play a prominent role in infections and inflammation [1, 2]. The dynamics and molecular mechanism of monocyte recruitment and the role of monocytes, particularly Ly6C$^{low}$ monocytes, during brain fungal infections remain largely unknown. In this study, with the use of IVM, we observed a remarkable influx of monocytes in the brain vasculature starting at 12 h post infection with *C. neoformans*. Interestingly, the Ly6C$^{low}$ subset comprised the majority of the monocytes recruited to the brain, although an influx of Ly6C$^{hi}$ monocytes was also noted. In this regard, recent data demonstrated that Ly6C$^{low}$ monocytes were remarkably recruited to the joint during the development of arthritis in mice and differentiated into inflammatory macrophages, promoting disease pathogenesis [13]. It was reported that Ly6C$^{low}$ monocytes recruited neutrophils in mesenteric veins, driving vascular inflammation [12]. In addition, Ly6C$^{low}$ monocytes were seen to interact with neutrophils through cell-cell contact in the vasculature of kidneys, promoting neutrophils activation and ROS generation [8, 11]. In contrast, the influx of neutrophils to the brain was limited at 24 h and 48 h post infection compared to monocytes in our experimental setting. Neutrophils are usually the first cells that are rapidly recruited to infection sites; however, the frequency of neutrophils was low in the brain vasculature even at earlier time points following brain infection with *C. neoformans* [39].

It is important to note that Ly6C$^{hi}$ monocytes have been implicated to play important roles during pulmonary infection of *C. neoformans* [40, 41]. CCR2/CCL2 axis is involved in human brain infections with *C. neoformans* [42, 43]. The relatively less recruitment of Ly6C$^{hi}$ monocytes to the brain observed in our experimental setting might be attributed to the delay of CCR2$^+$Ly6C$^+$ monocyte mobilization from the bone marrow. Indeed, accumulation of CD11b$^+$Ly6C$^+$ myeloid cells was observed in the brain of infected mice only starting on 14 days after intravenous infections of *C. neoformans* strain 52D [44]. In this regard, we found that an initial pulmonary infection followed by intravenous infection two weeks later significantly enhanced Ly6C$^{hi}$ monocyte recruitment to the brain (S8 Fig).

Leukocyte migration depends on adhesion molecules [2]. Identifying the adhesion molecules involved in Ly6C$^{low}$ monocyte recruitment to the brain would be of interest for therapeutic purposes in order to selectively block the trafficking of the cells. The adhesion molecules in monocyte trafficking have been extensively studied using in vitro systems [2, 15–17, 45]. However, relatively less is known about the adhesion molecules for monocyte recruitment in vivo. During *Leishmania major* infection, monocyte migration through inflamed dermal venules relies on interactions of PSGL-1 with P- and E-selectins, and of L-selectin with PNAd, while migration through lymph node high endothelial venules is dependent on L-selectin-PNAd interactions [46]. In the setting of atherosclerosis, P-selectin, E-selectin, and VCAM1 are

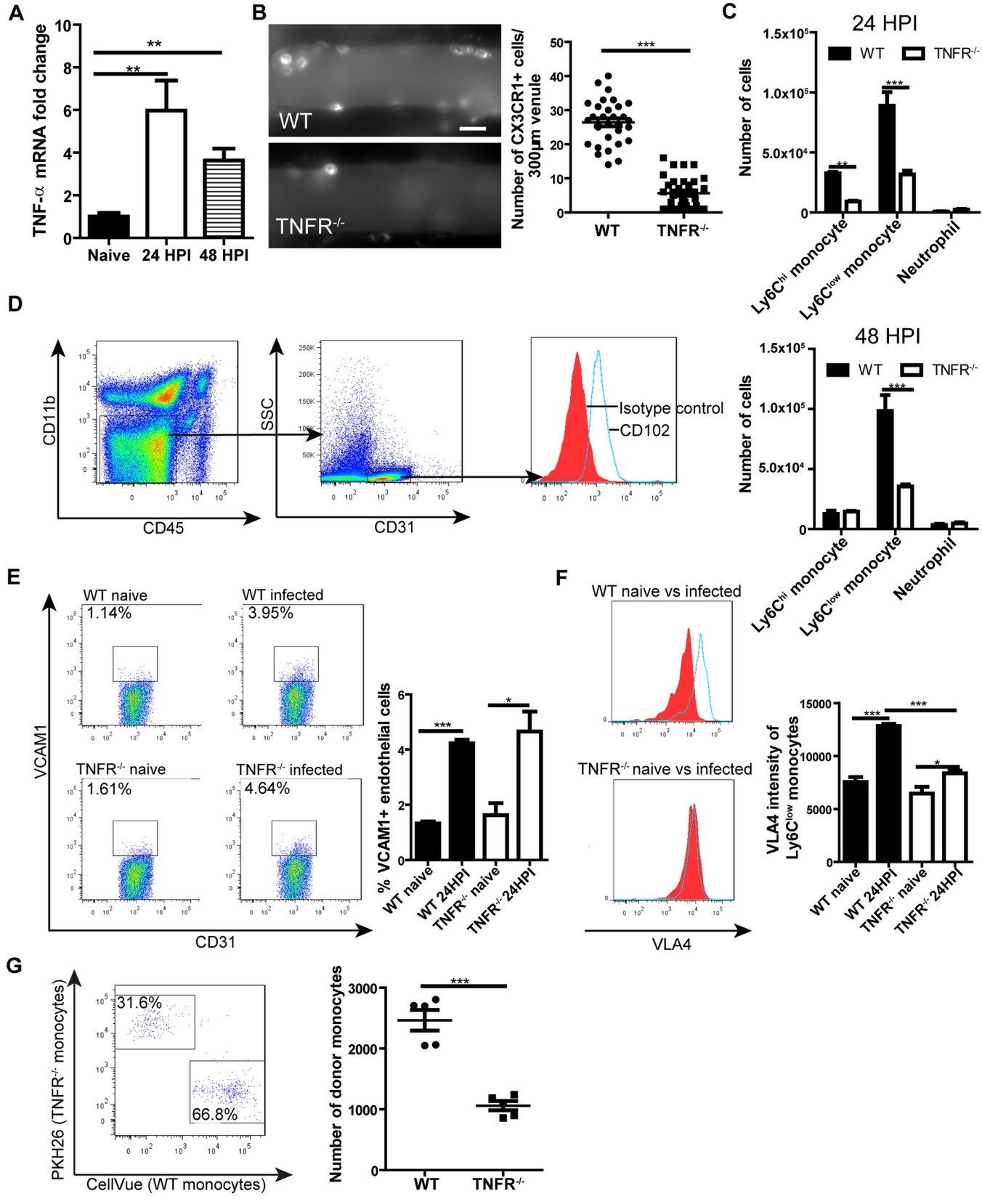

**Fig 5. TNFR signaling is crucial for monocyte recruitment to the brain by enhancing monocyte VLA4 expression. (A)** The level of TNF-α mRNA in the brain of mice (n = 5 per time point) before and after i.v. infection with 20x10^6 *C. neoformans*. **(B)** IVM analysis of monocyte recruitment to brain postcapillary venules of TNFR^-/- mice (n = 5) as compared to WT mice (n = 5) 24 h after i.v. infection with 20x10^6 *C. neoformans*. The infected mice were i.v. injected with 2 μg AF647-anti-CX3CR1 mAb to label monocytes 5 min before imaging. Left panel: representative images, right panel: quantification of monocytes. **(C)** Flow cytometry determination of the numbers of Ly6C^hi and Ly6C^low monocytes and neutrophils in the brain of TNFR^-/- and WT mice (n = 5 per group) 24 h (upper panel) and 48 h (lower panel) after i.v. infection with 20x10^6 *C. neoformans*. **(D)** The gating strategy for brain endothelial cells. Endothelial cells were defined as CD45^-CD11b^-CD31^+ population which demonstrated expression of CD102. **(E)** Flow cytometry determination of the percentage of brain endothelial cells expressing VCAM1 in WT and TNFR^-/- mice (n = 4 per group) 24 h after infection of 20x10^6 *C. neoformans* as compared to naïve mice. Left panel: representative plots, right panel: quantification. **(F)** Flow cytometry analysis of the expression of VLA4 on Ly6C^low monocytes from the brain of WT and TNFR^-/- mice (n = 5 per group) 24 h after infection with 20x10^6 *C. neoformans* as compared to naïve mice. Left panel: representative histograms, right panel: quantification. **(G)** Isolated monocytes from WT and TNFR^-/- mice (CD45.2 background) were stained by lipophilic dye CellVue and PKH26 respectively and mixed at 1:1 ratio. The mixed monocytes (2x10^6) were transferred into CD45.1 recipient mice (n = 5 mice) and the mice were i.v. infected with 20x10^6 *C. neoformans* for 24 h. Flow cytometry was performed to analyze the recruitment of adoptively transferred monocytes to the brain (left, gated on CD45.2^+CD45.1^- donor monocytes). The quantification of recruited donor monocytes was shown in the right panel. Scale bar: 10 μm. Data are expressed as mean ± SEM and representative of 2 independent experiments. *, p<0.05; **, p<0.01; ***, p<0.001.

involved in monocyte rolling on inflamed aortic endothelium [47]. Blocking CD11b or ICAM1 using antibodies inhibits Ly6C^hi monocyte recruitment to the liver during infection with *Listeria monocytogenes* [48]. P-selectin has been shown to mediate Ly6C^hi monocyte recruitment to the brain in a CCR2 dependent manner during peripheral organ inflammation [38, 49]. In the current study, we reveal that Ly6C^low monocyte recruitment to the brain was mainly mediated by the interaction of VCAM1 expressed on endothelial cells and VLA4 expressed on the monocytes following infection with *C. neoformans*. This is in contrast to steady condition in which crawling of Ly6C^low monocytes along endothelial cells of dermis is mediated by CD11a [7]. These results suggest that the adhesion mechanisms used by monocytes may differ, depending on the subset of monocytes, the type of tissues, the infectious agent, and the inflammatory state.

In this study, we have shown that TNFR signaling played an essential role in monocyte recruitment to the brain via enhancing VLA4 expression on monocytes. Previous studies have shown that intraperitoneal administration of TNF-α induced marked neutrophil rolling and adhesion in the brain microvasculature via activating brain endothelial cells and enhancing their expression of P- and E-selectin [37]. Accordingly, TNFR^-/- mice displayed significant reductions in rolling and adhesion of neutrophils in the brain in response to LPS [34]. These results demonstrated that TNFR signaling is critical for the neutrophil recruitment to the brain via acting on the brain endothelium. In contrast, in our experimental setting of brain infection with *C. neoformans* TNFR signaling mediated monocyte recruitment through acting on monocytes. Thus, although TNFR signaling is a central mediator of both neutrophil and monocyte recruitments, the underlying mechanisms appear distinct, reflecting the difference in adhesion molecules used by these cells for recruitment to the brain. Following *C. neoformans* infection, we detected enhanced expression of TNF-α in the brain. Microglia is a potential producer of TNF-α during brain inflammation [50]. In addition, it has been recently shown that Ly6C^low monocytes rapidly secrete a large amount of TNF-α during inflammation [11, 12] and infection with *L. monocytogenes* [7]. The cellular source of TNF-α in the brain infected with *C. neoformans* deserves further investigation.

*C. neoformans* can enter the brain via direct invasion of the brain endothelium [22–25]. There is also evidence that monocytes can act as Trojan horse, contributing to brain invasion by *C. neoformans* [23, 26, 27]. Notably, live cell imaging has been recently performed to directly visualize the interactions of monocytes containing *C. neoformans* with a monolayer of human brain endothelial cells cultured in vitro, leading to transmigration of the infected monocytes [30, 31]. Taking advantage of IVM, we have visualized the dynamic interactions among recruited monocytes, *C. neoformans* and brain vasculature in live animals. Following phagocytosis of *C. neoformans*, monocytes carrying the organism crawled on and adhered to

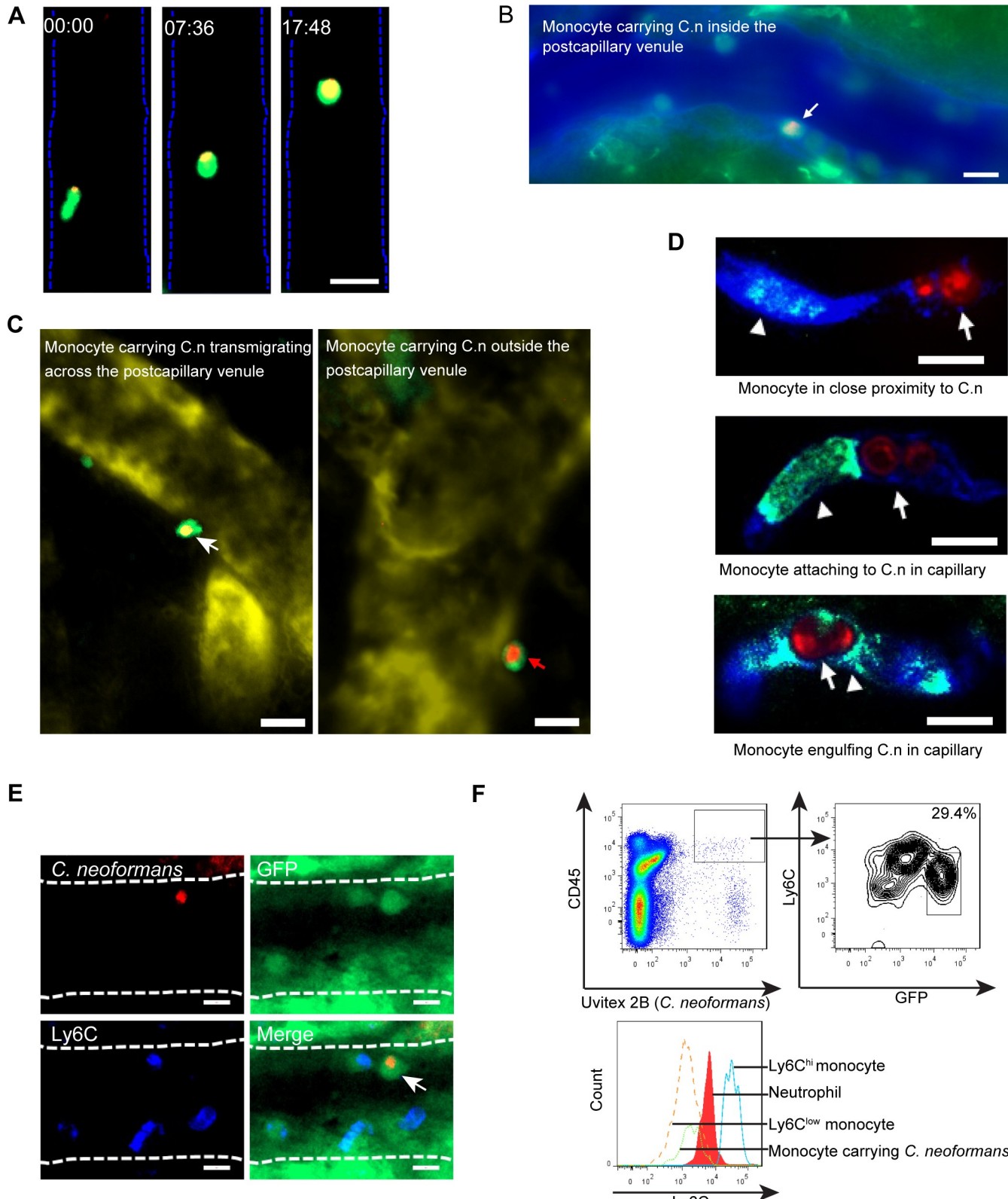

**Fig 6. Ly6C<sup>low</sup> monocytes engulf *C. neoformans* and carry the organism as they crawl on and adhere to the vessel wall and migrate to the brain parenchyma. (A)** A series of IVM representative images showing that a GFP<sup>+</sup> monocyte (green) carrying an ingested *C. neoformans* (red) was crawling on the luminal side of a postcapillary venule in the brain of infected CX3CR1<sup>gfp/+</sup> mice. IVM was performed on the brain of mice 18 h after infection with 20x10<sup>6</sup> *C.*

*neoformans*. See also S10 Video. **(B)** A representative IVM image showing that a GFP$^+$ monocyte (green) carrying *C. neoformans* (red) was attached to the luminal side of a postcapillary venule in the brain of infected CX3CR1$^{gfp/+}$ mice. IVM was performed 18 h post infection with 20x10$^6$ *C. neoformans* and the blood vessel (blue) was labeled with AF647-conjugated BSA. See also S11 Video. **(C)** Representative IVM images showing that GFP$^+$ monocytes (green) carrying *C. neoformans* (red) were in the process of crossing the vessel wall (left panel) or were located outside the postcapillary venule (right panel). IVM was performed on the brain in CX3CR1$^{gfp/+}$ mice 18 h post infection with *C. neoformans* and the blood vessel (yellow) was labeled with AF647-conjugated BSA. **(D)** Monocytes (green, arrowhead) were seen in close proximity to *C. neoformans* (red, arrow) in capillary vessels (blue, upper panel), or attaching to *C. neoformans* (middle panel), or engulfing *C. neoformans* within capillary vessels (lower panel). Immunohistochemistry was performed for the brain tissues of C57BL/6 mice 18 h after infection with 20x10$^6$ *C. neoformans*. **(E)** Representative IVM images showing GFP$^+$ monocytes (green) carrying *C. neoformans* (red, arrow) were Ly6C (blue, AF647-anti-Ly6C mAb) negative. IVM was performed on the brain of CX3CR1$^{gfp/+}$ mice 18 h after infection with *C. neoformans*. **(F)** Representative flow cytometry histograms showing the expression level of Ly6C on monocytes carrying *C. neoformans*. Brain leukocytes were purified from CX3CR1$^{gfp/+}$ mice (n = 5) 18 h after infection with 20x10$^6$ Uvitex 2B labeled *C. neoformans*. Monocytes carrying *C. neoformans* were defined as CD45$^+$Uvitex 2B$^+$GFP$^+$ cells (upper panel). The expression of Ly6C on monocytes carrying *C. neoformans* (CD45$^+$Uvitex 2B$^+$GFP$^+$) was compared to Ly6C expression by Ly6C$^{hi}$ monocytes, neutrophils, and Ly6C$^{low}$ monocytes (lower panel). Scale bars: 10 μm.

postcapillary venules and were occasionally seen to migrate to the brain parenchyma. Interestingly, the monocytes carrying *C. neoformans* were almost exclusively Ly6C$^{low}$ monocytes. Given that both Ly6C$^{hi}$ and Ly6C$^{low}$ monocytes have the potential to internalize *C. neoformans*, we cannot exclude the possibility of transition of Ly6C$^{hi}$ monocytes into Ly6C$^{low}$ monocytes following phagocytosis of the organism, as in situ monocyte transition (from Ly6C$^{hi}$ to Ly6C$^{low}$) has been previously documented [32, 33]. It is worthy to note that Ly6C$^{low}$ monocytes have been recently shown to carry ingested amyloid beta as they crawl on the luminal wall of brain vasculature [51]. In addition to postcapillary venules, we also visualized the behavior of monocytes in capillaries. Monocytes adopted a rod-shaped morphology when crawling within capillaries. Monocytes carrying *C. neoformans* were also observed in capillaries. It has been reported that rod-shaped Ly6C$^{low}$ monocytes patrol the brain capillaries and give rise to perivascular macrophages under inflammatory conditions [52]. It remains to be determined whether monocytes carrying *C. neoformans* in capillaries transmigrate to the brain parenchyma.

The transcription factor Nr4a1 controls Ly6C$^{low}$ monocytes differentiation and survival [53–55]. Consequently, Nr4a1$^{-/-}$ mice displayed dramatically diminished Ly6C$^{low}$ monocyte recruitment to the brain after fungal infection (S9A Fig). We hypothesized that the reduced Ly6C$^{low}$ monocyte recruitment in Nr4a1$^{-/-}$ mice led to a decreased fungal burden in the brain. We repeatedly observed a slight reduction of brain CFU (~ 15%) in infected Nr4a1$^{-/-}$ mice compared to infected wild-type mice (S9B Fig). Blocking VCAM1 or VLA4 also slightly reduced brain fungal burden (S9C Fig). However, the reduction of brain CFU did not reach statistical difference. ICAM1 deficiency enhanced brain CFU (S9D Fig), which may reflect less killing of *C. neoformans* in the brain vasculature due to reduced recruitment of neutrophils [56]. It is noteworthy that, in addition to hijacking of phagocytes, *C. neoformans* can use other mechanisms including transcytosis and paracytosis to invade the brain [57]. It is likely that Ly6C$^{low}$ monocytes make less contributions to *C. neoformans* invasion into the brain in our model system. Although Ly6C$^{low}$ monocytes were seen to carry *C. neoformans* while crawling on and adhering to the luminal wall of brain vasculature and migrating to the brain parenchyma, further investigation is required to determine the relative contribution of this subset compared to other mechanisms involved in brain invasion by *C. neoformans*.

In summary, there was a substantial recruitment of monocytes in the brain following infection with *C. neoformans*. Interestingly, the recruitment was primarily of Ly6C$^{low}$ monocytes and mainly mediated by the interaction of VCAM1 with VLA4 in our model system. TNFR signaling played an essential role in monocyte recruitment by enhancing VLA4 expression on monocytes. Although the conclusions were mainly made using high dose infection, we proved that the mechanisms also apply to low dose infection (S10 Fig). Finally, in addition to the previously described anti- and pro-inflammatory properties, we showed that Ly6C$^{low}$ monocytes

were capable of internalizing *C. neoformans* and carrying the ingested pathogen as they crawled along the luminal wall of the brain vasculature and migrated to the brain parenchyma.

# Materials and methods

## Ethics statement

This study was performed in strict accordance with the recommendations in the Guide for the Care and Use of Laboratory Animals from National Research Council. Animals study protocols were approved by the Institutional Animal Care and Use Committee (IACUC) of University of Maryland, College Park under identification number R-15-17 and R-SEPT-18-51.

## Animals

Wild-type C57BL/6 mice in CD45.2 background or CD45.1 background were purchased from the National Cancer Institute (Frederick, MD). CX3CR1^gfp/gfp, CCR2^rfp/rfp, CD11a^-/-, CD11b^-/-, ICAM1^-/-, Nr4a1^-/-, and TNFR^-/- mice in C57BL/6 background were obtained from Jackson Laboratory. CX3CR1^gfp/gfp mice were bred with wild-type C57BL/6 mice or CCR2^rfp/rfp mice to produce CX3CR1^gfp/+ mice or CX3CR1^gfp/+CCR2^rfp/+ heterozygous mice. The CX3CR1^gfp/gfp mice express GFP under the promoter of the CX3CR1 gene [58], which is predominantly expressed on monocytes and has been extensively used to study monocyte behavior [11, 12, 32]. All colonies were maintained in ventilated specific-pathogen-free facilities with standard 12 h light/dark cycles. Animals between 6 and 12 weeks of age were used for all experiments.

## *C. neoformans* and infection

The encapsulated *C. neoformans* strain H99 (serotype A) was obtained from the ATCC (Catalog# 208821). The GFP-expressing H99 strain was provided by Dr. Robin May (University of Birmingham). The tdTomato-expressing H99 strain was generated and provided by Dr. Xiaorong Lin (University of Georgia). In some experiments, the yeast cells were labeled by tetramethylrhodamine [25] or Uvitex 2B [59] before use. The organisms were grown to log phase in Sabouraud's dextrose broth at 32°C with gentle rotation overnight and washed twice in sterile PBS. For infection, mice were infected with $20 \times 10^6$ *C. neoformans* via the tail vein.

## Intravital microscopy

Intravital microscopy (IVM) was performed on the brain vasculature as previously described [25, 34]. Briefly, mice were anesthetized by i.p. injection of a mixture of 10 mg/kg xylazine and 200 mg/kg ketamine hydrochloride in 200 μl PBS. After confirming anesthesia, the skin over the skull was moistened with oil and removed using surgical scissors. A surgical craniotomy was then performed using a high-speed drill (Ideal Micro-Drill) with a tip diameter of 0.9 mm to generate a circle (5 μm in diameter) on the right side of the skull. The cranial window was opened by gently removing the covering skull to expose the underlying pial vasculature. After the surgery, the mouse was fixed on a customized stage for IVM. A drop of filter-sterilized artificial cerebrospinal fluid (119 mM NaCl, 26.2 mM NaHCO$_3$, 2.5 mM KCl, 1 mM NaH$_2$PO$_4$, 1.3 mM MgCl$_2$, 10 mM glucose, 2.5mM CaCl$_2$) was applied to the exposed portion of the brain and maintained throughout the experiment. The body temperature of the animals was maintained at 37°C throughout the experiment using a heating pad with a sensor probe inserted in the murine anus (Harvard apparatus). The FITC channel which detects GFP signal or tissue auto-fluorescence was first used to locate the field of view (FOV) of interest. Once the FOV was located, a video was captured through a 40x water immersion lens at a rate of 1 frame/

second for later analysis. A typical FOV of interest included a postcapillary venule of 30–70 μm in width and at least 300 μm in length as described previously [34]. In some experiments, mice were injected with 2 μg Alexa Fluor-647 conjugated anti-CX3CR1 or anti-Ly6C mAbs via the tail vein to label monocytes 5 min before visualization. To block the adhesion molecules, mice were injected with 100 μg blocking mAbs through the tail vein at indicated time.

## Video analysis

The numbers of rolling and adherent (including crawling) monocytes were quantified from IVM video analysis using the Zen software. The definition of rolling and adhesion followed previous publications [36, 60, 61]. Briefly, rolling monocytes were defined as those monocytes moving at a velocity slower than erythrocytes within a given vessel and showing discreet interaction with the vascular wall which interrupted their circulatory movements for less than 30 seconds. Adherent monocytes were defined as those monocytes continuously interacting with the blood vessel for more than 30 seconds. At a given time point, the total number as well as rolling/adherent monocytes (either GFP⁺ or CX3CR1⁺) within a blood vessel of 30–70 μm in width and 300 μm in length were enumerated.

## Brain leukocyte preparation

After anesthesia of the mice, blood was taken by cardiopuncture from the right ventricle of the mouse heart. Then, the inferior vena cava was cut open by surgical scissors, and 20 ml cold PBS was used to perfuse the body from the left ventricle to remove free blood cells in the circulation. After perfusion, the brain (including the cortex and cerebellum) was harvested into 15 ml tubes containing 2 ml RPMI 1640 medium, minced into small pieces with a 1 ml pipet tip. Collagenase IV (Worthington) was added to a final concentration of 1 mg/mL and the tissues were incubated for 30 min at 37˚C with gentle shaking. After enzymatic digestion, 100% Percoll solution was added to the tissue solution to a final concentration of 30% Percoll (GE), and filtered through a 70 μm cell strainer on ice. The tissues in 30% Percoll were layered onto a new 15 ml tube containing 2 ml 80% Percoll and centrifuged at 1500g for 15 min without brake. The leukocytes on the 80% and 30% interface were collected into a new 15 ml tube, washed with PBS; treated with ACK buffer for red blood cell lysis, and resuspended in 200 μl cold flow staining buffer (1% BSA in PBS with 0.05% sodium azide). Cells were counted under hemocytometer and ready for staining.

## Flow cytometry

For flow cytometry, up to 1x10⁶ brain leukocytes were suspended in 50 μl flow cytometry staining buffer. Fc receptors were blocked using anti-CD16/32 mAb (93; eBioscience) at 10 μg/ml for 20 min on ice. The cells were then stained by fluorophore conjugated mAbs for 30 min on ice. After washing, the cells were fixed in 1% PFA for 20 min; washed and resuspended in 200 μl flow staining buffer for detection using FACSCanto II flowcytometer (BD Biosciences). A minimum of 100000 events were recorded and the data obtained were analyzed using FlowJo software. Antibodies including anti-CD45 (30-F11, APC-Cy7), Ly6C (HK1.4, PERCP), CD45.1 (A20, APC-Cy7), CD45.2 (104, PE-Cy7), VLA4 (PS/2, PE), VCAM1 (429, AF647), NK1.1 (PK136, FITC) and Ly6G (1A8, AF647) were purchased from Biolegend.

## Immunohistochemistry

After infection, mice were euthanized, and the brain was removed after perfusion as described above. The brain was immediately frozen in OCT compound and cut on a cryostat microtome at a thickness of 5-μm sections onto coated glass slides. Tissue sections were fixed in ice cold acetone for 10 minutes. Sections were then incubated with 3% goat serum in PBS, followed by incubation with rabbit-anti-mouse collagen IV (Invitrogen), rat-anti-mouse F4/80 (eBioscience), and a mouse mAb specific for cryptococcal polysaccharide (E1, a gift from Françoise Dromer, Institut Pasteur, Paris) at 4°C overnight in a humidified chamber. After 3 washes, sections were incubated for 30 minutes with AF647 goat anti-rabbit IgG (Invitrogen) to delineate brain micro-vasculature, and with AF488 goat anti-rat IgG (Invitrogen) to identify monocytes, and AF555 goat anti-mouse IgG (Invitrogen) to stain the yeast cells in the brain microvasculature. The sections were rinsed and mounted with fluorescence anti-fading medium (KPL).

## Monocyte adoptive transfer

Monocytes used for adoptive transfer were isolated from the bone marrow of WT and TNFR$^{-/-}$ mice (both CD45.2 background) using monocyte isolation kit from Miltenyi Biotec (Catalog# 130-100-629) following manufacturer's instructions. The isolated monocytes from WT and TNFR$^{-/-}$ mice were separately labeled with lipophilic membrane dye CellVue Claret Far Red and PKH26 (both from Sigma-Aldrich) and mixed at 1:1 ratio. $2 \times 10^6$ mixed monocytes were transferred to each congenic mouse with CD45.1 background 2 h before infection with $20 \times 10^6$ *C. neoformans*; mice were euthanized 24 h post infection to examine the recruitment of donor monocytes.

## Quantitative PCR

A small piece of the brain was collected for RNA extraction by Trizol (Invitrogen) according to the manufacturer's instructions. The cDNA was synthesized by SuperScript IV First-Strand Synthesis System (Invitrogen). qPCR was performed on a Bio-Rad CFX96 qPCR instrument using SYBR Green PCR Master Mix (Applied Biosystems). The PCR primers for TNF and VCAM1 are as follows: VCAM1-F (AGT TGG GGA TTC GGT TGT TCT), VCAM1-R (CCC CTC ATT CCT TAC CAC CC). TNF-F (CAT CTT CTC AAA ATT CGA GTG ACA A), TNF-R (TGG GAG TAG ACA AGG TAC AAC CC).

## Statistics

Data were expressed as mean ± SEM. Evaluation of statistical significance was performed using Graph-Pad Prism 5 software (San Diego, CA). If only two groups were involved in the analysis, an unpaired two-sided Student's t-test was performed; for comparisons of more than two groups, one-way analysis of variance (ANOVA) was performed followed by Tukey's post-hoc test to determine significance among groups. In both cases, $p < 0.05$ is considered significant.

## Supporting information

**S1 Fig. A GFP$^+$ cell crawling against the blood flow along the brain postcapillary venule of CX3CR1$^{gfp/+}$ mice under steady conditions.** IVM was performed on the brain of a naïve CX3CR1$^{gfp/+}$ mouse. A series of images showing a monocyte (red arrow) crawling against the blood flow along the postcapillary venule (left panel). The enlargement on the right panel showing the leading edge of the crawling monocyte (white arrowhead). See also S2 Video.

Scale bars: 10 μm.
(TIF)

**S2 Fig. Depletion of monocytes abolishes GFP$^+$ cell recruitment to the brain vasculature of CX3CR1$^{gfp/+}$ mice infected with *C. neoformans*.** CX3CR1$^{gfp/+}$ mice (n = 5 per group) were i.v. administered with 200 μl Clodronate liposomes (CLL) to deplete monocytes or PBS liposomes as control. The mice were i.v. infected with 20x10$^6$ *C. neoformans* H99 24 h later. IVM was performed on the brain 24 h post infection to enumerate GFP$^+$ cells recruited to the brain postcapillary venules. Data are expressed as mean ± SEM. ***, p<0.001 by two-tailed student's *t* test.
(TIF)

**S3 Fig. Gating strategy of Ly6C$^{hi}$ and Ly6C$^{low}$ monocytes and analysis of GFP expression on NK cells in CX3CR1$^{gfp/+}$ mice.** (A) Ly6C$^{hi}$ monocytes were defined as CD45$^+$Ly6G$^-$NK1.1$^-$CD11b$^+$CX3CR1$^+$Ly6C$^{hi}$, while Ly6C$^{low}$ monocytes as CD45$^+$Ly6G$^-$NK1.1$^-$CD11b$^+$CX3CR1$^+$Ly6C$^{low}$. Microglia express intermediate level of CD45; they were gated out after selecting CD45$^+$ population. Leukocytes were isolated from the brain of C57BL/6 mice 24 h after i.v. infection with 20x10$^6$ *C. neoformans*. (B) A representative flow cytometry histogram showing the expression of GFP by CD45$^+$NK1.1$^+$ cells. CX3CR1$^{gfp/+}$ mice were i.v. infected with 20x10$^6$ *C. neoformans*. 24 h later, leukocytes were purified from the brain of infected mice for flow cytometry analysis.
(TIF)

**S4 Fig. IVM reveals limited neutrophil recruitment to the brain during *C. neoformans* infection.** C57BL/6 mice were i.v. infected with 20x10$^6$ GFP-labeled *C. neoformans*. A representative IVM image showing neutrophils in the brain of infected mice 24 h after infection. The mice were i.v. injected with 2 μg AF647-anti-Ly6G mAb to label neutrophils 5 min before imaging. *C. neoformans*: green, neutrophils: red. Scale bar: 10 μm.
(TIF)

**S5 Fig. Preferential accumulation of Ly6C$^{low}$ monocytes in the brain comparing to other organs after fungal infection.** (A) Mice (n = 5 per group) were i.v. infected with 20x10$^6$ *C. neoformans* H99. Twenty-four hours later, different organs were collected for enumeration of monocytes (Ly6C$^{hi}$ monocytes: CD45$^+$CD11b$^+$CX3CR1$^+$Ly6C$^{hi}$; Ly6C$^{low}$ monocytes: CD45$^+$CD11b$^+$CX3CR1$^+$Ly6C$^{low}$) by flow cytometry. (B) Representative IVM images showing the recruitment of CX3CR1$^+$ monocytes 24 h after i.v. infection with 20x10$^6$ *C. neoformans* H99 (red). Mice were i.v. injected with 5 μg AF647 conjugated anti-CX3CR1 mAb 10 min before imaging to label monocytes (blue). Scale bar 20 μm. Data expressed as mean ± SEM are representative of 2 independent experiments. *, p<0.05; ***, p<0.001 by two-way ANOVA.
(TIF)

**S6 Fig. Extravasation of circulating Ly6C$^{low}$ monocytes into the brain parenchyma.** Mice (n = 5 per group) were i.v. infected with 20x10$^6$ *C. neoformans* H99 for 1 or 2 days. Ten minutes before euthanasia, mice were i.v. treated with 5 μg PE-Cy7 conjugated anti-CD45.2 mAb (clone: 104), which does not block CD45 (clone: 30-F11) mAb binding, to label all circulatory leukocytes. Brain leukocytes were then isolated and analyzed by flow cytometry. (A) Representative plots showing the percentage of transmigrated Ly6C$^{low}$ monocytes (CD45.2$^-$, outside brain blood vessels) out of the total brain CD45$^+$ leukocytes. (B) The percentage (left) and number (right) of transmigrated Ly6C$^{low}$ monocytes over the time. Data are expressed as mean ± SEM. ***, p<0.001 by one-way ANOVA followed by Tukey's test.
(TIF)

**S7 Fig. Characterization of the interactions of monocytes carrying *C. neoformans* with brain vasculature during *C. neoformans* infection.** (A) A series of IVM images showing that a GFP$^+$ monocyte (green) carrying *C. neoformans* (red) was crawling in the luminal side of a postcapillary venule of CX3CR1$^{gfp/+}$ mice 18 h post i.v. infection with 20x10$^6$ *C. neoformans*. (B) Immunohistochemistry showing a monocyte containing multiple *C. neoformans*, one of which appeared to spread from the monocyte to an endothelial cell. C57BL/6 mice were infected with 20x10$^6$ *C. neoformans* and euthanized 18 h after infection for immunohistochemistry. Upper panel: 2D images; lower panel: 3D image. Monocytes: green, *C. neoformans*: red, vessel: blue. (C) The percentages of phagocytes carrying *C. neoformans*. CX3CR1$^{gfp/+}$ mice (n = 5) were infected with 20x10$^6$ Uvitex 2B labeled *C. neoformans*. Brain leukocytes were purified 18 h post infection for flow cytometry analysis. Initially, CD45$^+$Uvitex 2B$^+$ population were gated. The percentages of monocytes carrying *C. neoformans* (Ly6G$^-$GFP$^+$) and neutrophils carrying *C. neoformans* (Ly6G$^+$GFP$^-$) were analyzed. Upper left panel: a representative plot, upper right panel: quantification, lower panel: the percentage of free yeast cells in the brain. Scale bars: 10 μm. Data are expressed as mean ± SEM and representative of 2 independent experiments. $^{***}$, p<0.001 by two-tailed student's *t* test.
(TIF)

**S8 Fig. Pulmonary infection (i.n. infection) enhances Ly6C$^{hi}$ monocyte recruitment to the brain, associated with reduced Ly6C$^{low}$ monocyte recruitment.** Mice (n = 5 per group) were intranasally (i.n.) infected with 1x10$^4$ *C. neoformans* H99. Two weeks later, the infected mice (i.n. & i.v. infection group) and uninfected control mice (i.v. infection group) were infected with 20x10$^6$ *C. neoformans* H99 through the tail vein. Twenty-four hours later, Ly6C$^{hi}$ and Ly6C$^{low}$ monocytes in the brain were analyzed by flow cytometry. Left: representative plots showing Ly6C$^{hi}$ and Ly6C$^{low}$ monocytes. Right: quantification of Ly6C$^{hi}$ and Ly6C$^{low}$ monocytes. Data expressed as mean ± SEM are representative of 2 independent experiments. $^{***}$, p<0.001 by two-way ANOVA.
(TIF)

**S9 Fig. Reduced Ly6C$^{low}$ monocyte recruitment in the brain of infected Nr4a1$^{-/-}$ mice and analysis of brain fungal burdens.** (A) WT mice and Nr4a1$^{-/-}$ mice (n = 5 per group) were i.v. infected with 20x10$^6$ *C. neoformans* H99 for 24 h; brain monocytes were analyzed by flow cytometry. Left: representative plots showing Ly6C$^{hi}$ and Ly6C$^{low}$ monocytes in the brain; right: the number of Ly6C$^{hi}$ and Ly6C$^{low}$ monocytes in the brain. (B) Brain fungal burdens of WT and Nr4a1$^{-/-}$ mice (n = 5–6 per group) 48 h after i.v. infection with 5x10$^4$ *C. neoformans* H99. (C) Brain fungal burdens of WT mice 48 h after i.v. infection with 5x10$^4$ *C. neoformans* H99. Mice (n = 5 per group) were treated with 100 μg anti-VCAM1, anti-VLA4 antibody or control antibody on day 0. (D) Brain fungal burdens of WT, CD11a$^{-/-}$, ICAM1$^{-/-}$ (n = 5 per group) 48 h after i.v. infection with 5x10$^4$ *C. neoformans* H99. Data are expressed as mean ± SEM. $^*$, p<0.05; $^{***}$, p<0.001 by two-tailed student's *t* test.
(TIF)

**S10 Fig. Monocyte recruitment to the brain during infection with a low dose of *C. neoformans*.** (A) Wild-type mice (n = 5 per group) were i.v. infected with 5x10$^4$ *C. neoformans* H99. Twenty-four hours later, infected and naïve mice were euthanized and the numbers of leukocytes in the brain were enumerated by flow cytometry. (B) TNFR$^{-/-}$ mice (n = 5 per group) were i.v. infected with 5x10$^4$ *C. neoformans* H99. The numbers of leukocytes in the brain of infected mice and naïve mice were counted 24 h after infection by flow cytometry. (C) Wild-type mice (n = 5 per group) were i.v. infected with 5x10$^4$ *C. neoformans* H99 for 24 h. 20 min before euthanasia, mice were treated with anti-VCAM1 or anti-VLA4 mAb. The numbers of

Ly6C$^{hi}$ and Ly6C$^{low}$ monocytes were enumerated by flow cytometry. Data are expressed as mean ± SEM. **, p<0.01; ***, p<0.001 by two-way ANOVA.
(TIF)

**S1 Video. IVM video showing that a GFP$^+$ monocyte was crawling inside the postcapillary venule in CX3CR1$^{gfp/+}$ mice under naïve conditions.**
(AVI)

**S2 Video. IVM video showing that a GFP$^+$ monocyte was crawling against the blood flow inside the postcapillary venule in CX3CR1$^{gfp/+}$ mice under naïve conditions.**
(AVI)

**S3 Video. IVM video showing the substantial recruitment of GFP$^+$ monocytes in the postcapillary venule of CX3CR1$^{gfp/+}$ mice 24 h after i.v. infection with 20x10$^6$ *C. neoformans*.**
(AVI)

**S4 Video. IVM video showing the recruitment of RFP$^+$ monocytes to the brain postcapillary venule in CCR2$^{rfp/rfp}$ mice 24 h after i.v. infection with 20x10$^6$ *C. neoformans*.**
(AVI)

**S5 Video. IVM video showing the recruitment of CX3CR1$^+$ monocytes to the brain postcapillary venule in C57BL/6 mice 24 h after i.v. infection with 20x10$^6$ *C. neoformans*.** Monocytes were labeled by i.v. injection of 2 µg AF647-anti-CX3CR1 mAb 5 min before imaging.
(AVI)

**S6 Video. Anti-VLA4 antibody treatment wipes off monocytes from the brain vasculature.** CX3CR1$^{gfp/+}$CCR2$^{rfp/+}$ mice were i.v. infected with 20x10$^6$ *C. neoformans*. IVM was performed on the brain 24 h after infection. During the imaging, mice were i.v. injected with 100 µg anti-VLA4 mAb to block VLA4.
(AVI)

**S7 Video. IVM video showing the recruitment of CX3CR1$^+$ monocytes to the brain postcapillary venule in CD11a$^{-/-}$ mice 24 h after i.v. infection with 20x10$^6$ *C. neoformans*.** Monocytes were labeled by i.v. injection of 2 µg AF647-anti-CX3CR1 mAb 5 min before imaging.
(AVI)

**S8 Video. IVM video showing the recruitment of CX3CR1$^+$ monocytes to the brain postcapillary venule in TNFR$^{-/-}$ mice 24 h after i.v. infection with 20x10$^6$ *C. neoformans*.** Mice were i.v. administered with 2 µg AF647-anti-CX3CR1 mAb to label monocytes 5 min before imaging.
(AVI)

**S9 Video. IVM video showing that a monocyte (green) carrying *C. neoformans* (red) was crawling along the luminal side of the brain postcapillary venule in CX3CR1$^{gfp/+}$ mice 18 h after i.v. infection with 20x10$^6$ *C. neoformans*.**
(AVI)

**S10 Video. IVM video showing that a monocyte (green) carrying *C. neoformans* (red) was adhering to the luminal side of the brain postcapillary venule in CX3CR1$^{gfp/+}$ mice 18 h after i.v. infection with 20x10$^6$ *C. neoformans*.** The vasculature (blue) was labeled by AF647-BSA.
(AVI)

**S11 Video. IVM video showing that a monocyte (green) was crawling in the brain capillary and adopting a rod-shaped morphology when passing through the capillary in CX3CR1$^{gfp/+}$ mice 18 h after infection with 20x10$^6$ *C. neoformans.***
(AVI)

## Acknowledgments

Special thanks go to Dr. Youbao Zhao and Dr. Xiaorong Lin (University of Georgia) for the generation of tdTomato-expressing *C. neoformans*. We also thank Kenneth Class and Dr. Yunsheng Wang (University of Maryland College Park) for their assistance with FACS analyses.

## Author Contributions

**Conceptualization:** Donglei Sun, Mingshun Zhang, Gongguan Liu, Yanli Chen, Yong Fu, Meiqing Shi.

**Data curation:** Donglei Sun, Mingshun Zhang, Peng Sun.

**Formal analysis:** Donglei Sun, Mingshun Zhang.

**Funding acquisition:** Meiqing Shi.

**Investigation:** Donglei Sun, Mingshun Zhang, Peng Sun, Mohammed Yosri.

**Methodology:** Donglei Sun, Mingshun Zhang.

**Supervision:** Meiqing Shi.

**Validation:** Ashley B. Strickland, Meiqing Shi.

**Writing – original draft:** Donglei Sun, Mingshun Zhang, Meiqing Shi.

**Writing – review & editing:** Donglei Sun, Mingshun Zhang, Ashley B. Strickland, Meiqing Shi.

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
