## [Decision Letter · Decision Letter 0]

4 Oct 2019

Dear Dr. Shi,

Thank you very much for submitting your manuscript "VCAM1/VLA4 interaction mediates Ly6Clow monocyte recruitment to the brain in a TNFR signaling dependent manner during fungal infection" (PPATHOGENS-D-19-01682) for review by PLOS Pathogens. Your manuscript was fully evaluated at the editorial level and by independent peer reviewers. The reviewers appreciated the attention to an important problem, but raised some substantial concerns about the manuscript as it currently stands. In particular, reviewers raised concerns regarding the clinical relevance of the model and the role of Ly6Clow monocytes during infection, as well as issues related to discrepancies in findings presented in your manuscript with those in the literature.  In addition, reviewers requested additional experiments to address these concerns. These issues must be addressed before we would be willing to consider a revised version of your study. We cannot, of course, promise publication at that time.

We therefore ask you to modify the manuscript according to the review recommendations before we can consider your manuscript for acceptance. Your revisions should address the specific points made by each reviewer.

(1) A letter containing a detailed list of your responses to the review comments and a description of the changes you have made in the manuscript. Please note while forming your response, if your article is accepted, you may have the opportunity to make the peer review history publicly available. The record will include editor decision letters (with reviews) and your responses to reviewer comments. If eligible, we will contact you to opt in or out.

(2) Two versions of the manuscript: one with either highlights or tracked changes denoting where the text has been changed; the other a clean version (uploaded as the manuscript file).

Additionally, to enhance the reproducibility of your results, PLOS recommends that you deposit your laboratory protocols in protocols.io, where a protocol can be assigned its own identifier (DOI) such that it can be cited independently in the future. For instructions see http://journals.plos.org/plospathogens/s/submission-guidelines#loc-materials-and-methods

We hope to receive your revised manuscript within 60 days. If you anticipate any delay in its return, we ask that you let us know the expected resubmission date by replying to this email. Revised manuscripts received beyond 60 days may require evaluation and peer review similar to that applied to newly submitted manuscripts.

[LINK]

Sincerely,

Mairi C. Noverr, Ph.D.

Associate Editor

PLOS Pathogens

Aaron Mitchell

Section Editor

PLOS Pathogens

Kasturi Haldar

Editor-in-Chief

PLOS Pathogens

orcid.org/0000-0001-5065-158X

Grant McFadden

Editor-in-Chief

PLOS Pathogens

orcid.org/0000-0002-2556-3526

Reviewer's Responses to Questions

**Part I - Summary**

Reviewer #1: This intravital microscopy study, combined with analysis of CNS immune response responses focuses on the of Ly6Clow monocytes during the early stages of CNS disseminating fungal infection caused by C. neoformans (C.n). Study demonstrates rapid/early recruitment to the brain vasculature (within 12-24h) of monocytes, chielfly LY6Clow subset, following intravenous inoculation with a high-dose of C.n. H99. The recruitment depends primarily on the interaction of VCAM1 expressed on the brain endothelium with VLA4 expressed on Ly6Clow monocytes. TNFR signaling is essential for the recruitment through enhancing VLA4 expression on Ly6Clow monocytes. Some of these Ly6Clow monocytes contain internalized C. neoformans and carried the organism while crawling on and adhering to the luminal wall of brain vasculature and migrating to the brain parenchyma. This study highlights important involvement of this subset during infection. While focus on the Ly6Clow monocytes in the context of disseminated C.n., followed by CNS infection is novel and interesting and could potentially be an important advance, there are some significant concerns:

First, the presented results are difficult to reconcile with previous studies showing that CCR2/CCL2 axis is involved in human patients with CNS C.n. infection (Jarvis et all 2015; Panackal et al 2015) and CD11b+Ly6C+ myeloid cells are recruited in high quantities into the brain of mice with CNS C.n. infection (Neal at al 2017). These studies have not been cited or discussed by the authors. However, these published data STRONGLY argue against proposed here "unique link" of Ly6Clow monocytes with cryptococcal brain infection, as opposed to other organs with C.n-infection or the CNS infections with other microbes. The more likely reason for predominance of Ly6Clow monocytes in this model is that the relatively early time points (1-3 days) are used prior to CCR2+ monocyte BM mobilization, and the known delay in recruitment of Ly6Chi/CCR2+ monocytes, typically past day 7 and day 14 in the C.n-infected lungs or CNS, respectively.

Second, it is not clear what fraction of these intravascular monocytes actually crosses to the brain (as implied by the title and several paragraph titles). Thus, at this point, it is hard to understand relative importance of these monocyte subsets in host defenses/pathogenesis of the C.n-induced CNS disease without any quantitative data in this respect.

Reviewer #2: Overall, the studies were well executed and the findings/interpretations are supported by the data presented. Additionally, the article is well-written. The strengths of the study is that the authors findings were supported using multiple animal models and good visuals. Nonetheless, a weakness of the study is a concern that the infectious dose combined with the route of inoculation used in the study (20 x 10^6 i.v.) is so over-bearing on the immune response that the resulting observations are not relatable clinically. The overall consensus is that infection begins via intranasal inhalation and that the organisms may eventually disseminate to the CNS (typically in an immune suppressed host but not entirely). This allows for the development of a much different type of immune response and possibly more time for monocytes to differentiate into various other cell types such as DCs. Thus, the findings presented in the manuscript may only be a reflection of the model system. This may need to be addressed to put the observations in context or some additional studies done using a much smaller infectious dose and/or route of infection.

Reviewer #3: This is a study by Sun et al that defines the kinetics of accumulation of Ly6c low monocytes in the brain of Cryptococcus-infected mice and examines molecular factors that are associated with these cells' recruitment into the tissue. The authors utilize intravital microscopy to visualize some of these events and show that VCAM1/VLA4 and TNF receptor signaling is important for Ly6c low accumulation in the context of cryptococcal meningitis. This is a potentially interesting topic, with description of a cell type that has not been examined thoroughly during infections, and particularly during fungal infections. The authors carefully perform experiments to outline the molecules that may be critical for cell accumulation in the fungal-infected brain. The paper is well-written. However, the manuscript lacks 2-3 very important pieces of information to provide it with the significance in advancing the field as outlined below.

**Part II – Major Issues: Key Experiments Required for Acceptance**

Reviewer #1: The key experiment to address the first concern would be to monitor the vasculature of other organs (e.g. kidney and liver) in this C.n high-dose IV bolus disseminated infection model, to determine if similar microvasculature recruitment of Ly6Clow versus Ly6Chigh monocytes will be observed outside of the CNS in other organs during fungemia.

It would be also worthwhile to induce an inhalation C.n infection 1-2 week prior to the IV-bolus infection to activate bone marrow to produce both subsets of monocytes first, and then follow with the IV infection/microscopy studies. Such model would be more consistent with natural history of cryptococcal infection, when the infection initially develops in the lungs, triggers some level of host response, but then disseminates into the CNS. This approach would allow to verify the relative contribution of each monocyte subsets in the CNS in a more "natural history disease setting" and allow for better comparisons with other studies (as they all focus on less acute time points).

The second major concern regarding true monocyte recruitment to the brain could be addressed using short-term injection with fluorescent antibody or other monocyte tracer IV infusion (such as pkH20) at the onset of monocyte vascular accumulation and compare with similar approach at day 3 after infection and flow cytometry study to determine what fractions of Ly6C low monocles remained in the vasculature (tracer positive) and what fraction crossed BBB (tracer negative) over time. This study would help to clarify the claim of monocyte recruitment to the brain (as stated in the title of the manuscript).

If authors cannot produce the requested data the conclusions should be significantly toned down, limitations of the study acknowledged and alternative interpretations of their data should be provided.

Reviewer #2: 1. The authors should perform studies using a much lower infectious dose via the i.v. to show the necessity of VCAM1/VLA4 interactions and TNFR signaling on the recruitment of Ly6C lo monocytes to the brain. Doing these studies via the pulmonary route of infection may not be feasible using this strain of C. neoformans (H99) b/c mice typically succumb to infection via asphyxiation.

Reviewer #3: 1. Although the authors describe accumulation of Ly6c low monocytes in the fungal-infected brain and define some molecules important for this process, there is absolutely no information with regard to whether Ly6c low monocytes are protective or detrimental during infection or whether their accumulation plays no role. Therefore, the authors should perform two sets of experiments to enlighten the readers on this very important topic:

1a- Is Cryptococcus fungal load greater in any of the models that the ly6c low monocyte accumulation is low? That is in mice with impaired VCAM1 or VLA4 or CD11a or ICAM1?

1b- Is Cryptococcus fungal burden increased in Nr4a1-deficient mice that have been reported to specifically lack Ly6c low monocytes? (Carlin et al., Cell, 2013; Thomas et al., Immunity, 2016).

These 2 sets of experiments are in this reviewer's opinion critical to establish a functional role of Ly6c low accumulation in this model. Without that, it is unclear what the significance of modest decreases of Ly6c low monocytes in this model mean.

2. In Figure 5C, are the authors sure that the CD31+ cells are indeed endothelial cells and that the staining is not an artifact? Have they used CD102 to also stain? FACS sorted the cells to confirm by pPCR or WB that indeed these are endothelial cells? Brain preparations don't typically contain many endothelial cells unless specific protocols are followed, and such protocols are not specified in the methods by the authors. To extend this, the data on TNFR expression by these CD31+ cells in Figure 5E is quite unimpressive. The authors should use a positive control Ab that is known to be expressed in endothelial cells to prove that these are indeed valid data.

3. In Figure 5G the authors "normalize" numbers of Ly6c low monocytes relative to microglia. This is not the correct way to do this analysis. Absolute numbers of Ly6c low monocytes is important to show. In addition, these chimera data are suggestive but not proving of the point. Competitive repopulation studies of Ly6clow monocytes from 50% WT and 50% TNFR KO mice (1:1 mix, using different congenic markers) is the way to show that there is differential recruitment/accumulation. Therefore, the authors should not "normalize" numbers of monocytes to microglia and they should perform a competitive repopulation study to conclusively examine their hypothesis.

**Part III – Minor Issues: Editorial and Data Presentation Modifications**

Reviewer #1: Ln 91-92: the 600,000 numbers come from the older reports from the peak years of infections. Present estimate is around 180,000 deaths per year. please use more current reference such as the recent colloquium report of Academy of ASM

Ln 108-9 "following brain infection": to be precise the study is not restricted to brain infection but follows the acute fungemia, a model of acutely disseminated C.n. infection.

Ln: 117 the paragraph title is somewhat misleading since it is not clear from the study if the monocytes were recruited to the brain, brain vasculature or the vasculature of all other tissues

LN 138 In the absence of additional data it is not clear which subset of monocytes predominantly was recruited/crossed to the brain

Other titles Ln 167, 189, 206 should be probably changed from "to the brain" to "to the brain vasculature", as most images seem to show monocytes in the vasculature.

Since microglia express CX3CR1, the author should how they gate out microglia in their flow cytometry analysis such as Fig 2C. Does the CXCR3+Ly6Clow gate also contain microglia if you initially gated on CD45+ cells?

It is somewhat arbitrary what is "massive" cell recruitment to the brain (LN 40, 117, 136, 341, 661, 663, 848) but total cell numbers reported here are not that overwhelming in numbers. Perhaps this could be toned down to a "substantial".

Reviewer #2: 1. It is not clear by the monocyte population is compared to microglia (Figure 5G).

2. Do the authors suspect a difference in morbidity/mortality will be observed by blocking VCAM1/VLA4 interactions and/or TNFR signaling (perhaps due to less inflammatory cells trafficking to the brain)?

3. Adhesion misspelled in Y axis of Figure 4C.

Reviewer #3: 1. Why did the authors use VLA4 and VCAM1 Abs and not genetic deletion of these molecules? Are these mice not available?

2. Please show CD11b in Figure 2C. Are all CX3CR1+ cells (Ly6c hi and low) all CD11b+?

3. Please label Figure 6 appropriately (is #7 now)

PLOS authors have the option to publish the peer review history of their article (what does this mean?). If published, this will include your full peer review and any attached files.

Reviewer #1: No

Reviewer #2: No

Reviewer #3: No

---

## [Editor Report · Decision Letter 1]

28 Jan 2020

Dear Dr. Shi,

We are pleased to inform you that your manuscript 'VCAM1/VLA4 interaction mediates Ly6Clow monocyte recruitment to the brain in a TNFR signaling dependent manner during fungal infection' has been provisionally accepted for publication in PLOS Pathogens.

Before your manuscript can be formally accepted you will need to complete some formatting changes, which you will receive in a follow up email. A member of our team will be in touch within two working days with a set of requests.

Best regards,

Mairi C. Noverr, Ph.D.

Associate Editor

PLOS Pathogens

Aaron Mitchell

Section Editor

PLOS Pathogens

Kasturi Haldar

Editor-in-Chief

PLOS Pathogens

orcid.org/0000-0001-5065-158X

Michael Malim

Editor-in-Chief

PLOS Pathogens

orcid.org/0000-0002-7699-2064
---

## [Editor Report · Acceptance letter]

21 Feb 2020

Dear Dr. Shi,

We are delighted to inform you that your manuscript, "VCAM1/VLA4 interaction mediates Ly6Clow monocyte recruitment to the brain in a TNFR signaling dependent manner during fungal infection," has been formally accepted for publication in PLOS Pathogens.

Best regards,

Kasturi Haldar

Editor-in-Chief

PLOS Pathogens

orcid.org/0000-0001-5065-158X

Michael Malim

Editor-in-Chief

PLOS Pathogens

orcid.org/0000-0002-7699-2064